# On Intriguing Layer-Wise Properties of Robust Overfitting in Adversarial Training

## Abstract

Adversarial training has proven to be one of the most effective methods to defend against adversarial attacks. Nevertheless, robust overfitting is a common obstacle in adversarial training of deep networks. There is a common belief that the features learned by different network layers have different properties, however, existing works generally investigate robust overfitting by considering a DNN as a single unit and hence the impact of different network layers on robust overfitting remains unclear. In this work, we divide a DNN into a series of layers and investigate the effect of different network layers on robust overfitting. We find that different layers exhibit distinct properties towards robust overfitting, and in particular, robust overfitting is mostly related to the optimization of latter parts of the network. Based upon the observed effect, we propose a *robust adversarial training* (RAT) prototype: in a minibatch, we optimize the front parts of the network as usual, and adopt additional measures to regularize the optimization of the latter parts. Based on the prototype, we designed two realizations of RAT, and extensive experiments demonstrate that RAT can eliminate robust overfitting and boost adversarial robustness over the standard adversarial training.

## 1 Introduction

Deep neural networks (DNNs) have been widely applied in multiple fields, such as computer vision He et al. (2016) and natural language processing Devlin et al. (2018). Despite its achieved success, recent studies show that DNNs are vulnerable to adversarial examples. Well-constructed perturbations on the input images that are imperceptible to human's eyes can make DNNs lead to a completely different prediction Szegedy et al. (2013). The security concern due to this weakness of DNNs has led to various works in the study of improving DNNs robustness against adversarial examples. Across existing defense techniques thus far, Adversarial Training (AT) Goodfellow et al. (2014); Madry et al. (2017), which optimizes DNNs with adversarially perturbed data instead of natural data, is the most effective approach Athalye et al. (2018). However, it has been shown that networks trained by AT technique do not generalize well Rice et al. (2020). After a certain point in AT, immediately after the first learning rate decay, the robust test accuracy continues to decrease with further training. Typical regularization practices to mitigate overfitting such as l1 & l2 regularization, weight decay, data augmentation, etc. are reported to be as inefficient compared to simple early stopping Rice et al. (2020).

Many studies have attempted to improve the robust generalization gap in AT, and most have generally investigated robust overfitting by considering DNNs as whole. However, DNNs trained on natural images exhibit a common phenomenon: features obtained in the first layers appear to be general and applicable widespread, while features computed by the last layers are dependent on a particular dataset and task Yosinski et al. (2014). Such behavior of DNNs sparks a question: Do different layers contribute differently to robust overfitting? Intuitively, robust overfitting acts as an unexpected optimization state in adversarial training, and its occurrence may be closely related to the entire network. Nevertheless, the unique effect of different network layers on robust overfitting is still unclear. Without a detailed understanding of the layer-wise mechanism of robust overfitting, it is difficult to completely demystify the exact underlying cause of the robust overfitting phenomenon.

In this paper, we provide the first layer-wise diagnosis of robust overfitting. Specifically, instead of considering the network as a whole, we treat the network as a composition of layers and systematically investigate the impact of robust overfitting phenomenon on different layers. To do this, we first fix the parameters for the selected layers, leaving them unoptimized during AT, and then normally optimize other layer parameters. We discovered that robust overfitting is always *mitigated* in the case where the latter layers are left unoptimized, and applying the same effect to other layers is *futile* for robust overfitting, suggesting a strong connection between the optimization of the latter layers and the overfitting phenomenon.

Based upon the observed effect, we propose a *robust adversarial training* (RAT) prototype to relieve the issue of robust overfitting. Specifically, RAT works in each mini-batch: it optimizes the front layers as usual, and for the latter layers, it implements additional measures on these parameters to regularize their optimization. It is a general adversarial training prototype, where the front and latter network layers can be separated by some simple test experiments, and the implementation of additional measures to regularize network layer optimization can be versatile. For instance, we designed two representative methods for the realizations of RAT: $\text{RAT}_{\text{LR}}$ and $\text{RAT}_{\text{WP}}$. They adopt different strategies to hinder weight update, e.g., enlarging the learning rate and weight perturbation, respectively. Extensive experiments show that the proposed RAT prototype effectively eliminates robust overfitting. The contributions of this work are summarized as follows:

- We provide the first diagnosis of robust overfitting on different network layers, and find that there is a strong connection between the optimization of the latter layers and the robust overfitting phenomenon.

- Based on the observed properties of robust overfitting, we propose the RAT prototype, which adopts additional measures to regularize the optimization of the latter layers and is tailored to prevent robust overfitting.

- We design two different realizations of RAT, with extensive experiments on a number of standard benchmarks, verifying its effectiveness.

## 2 Related Work

### 2.1 Adversarial Training

Since the discovery of adversarial examples, there have been many defensive methods attempted to improve the DNN's robustness against such adversaries, such as adversarial training Madry et al. (2017), defense distillation Papernot et al. (2016), input denoising Liao et al. (2018), gradient regularization Tramèr et al. (2018). So far, adversarial training Madry et al. (2017) has proven to be the most effective method. Adversarial training comprises two optimization problems: the inner maximization and outer minimization. The first one constructs adversarial examples by maximizing the loss and the second updates the weight by minimizing the loss on adversarial data:

$$\ell^{\text{AT}}(\boldsymbol{w}) = \min_w \sum_i \max_{d(x_i, x_i') \leq \epsilon} \ell(f_w(x_i'), y_i), \tag{1}$$

where $f_w$ is the DNN classifier with weight $w$, and $\ell(\cdot)$ is the loss function. $d(.,.)$ specify the distance between original input data $x_i$ and adversarial data $x_i'$, which is usually an $l_p$-norm ball such as the $l_2$ and $l_\infty$-norm balls and $\epsilon$ is the maximum perturbation allowed.

A different type of AT variation that is commonly used is referred to as TRADES Zhang et al. (2019), which involves optimizing a surrogate loss that is a tradeoff between the natural accuracy and adversarial robustness:

$$\ell^{\text{TRADES}}(\boldsymbol{w}) = \min_w \sum_i \left\{ \text{CE}(f_w(x_i), y_i) \quad + \beta \cdot \max_{d(x_i, x_i') \leq \epsilon} \text{KL}(f_w(x_i) || f_w(x_i')) \right\}, \tag{2}$$

The surrogate loss consists of two parts: cross-entropy (CE) loss, which encourages the network to maximize natural accuracy, and Kullback-Leibler (KL) divergence, which encourages the improvement of robust accuracy. The hyperparameter $\beta$ is used to control the tradeoff between natural accuracy and adversarial robustness.

## 2.2 Semi-supervised learning

Another line of work involves utilizing semi-supervised learning (SSL) technique. Methods based on SSL use additional unlabeled data to improve the robustness of the trained model. In these methods, a natural model is first trained on labeled data to generate pseudo-labels for the unlabeled data. Then, a robust model is trained using an adversarial loss function $\ell(\boldsymbol{w})$ on both labeled and unlabeled data:

$$\ell^{\mathrm{SSL}}(\boldsymbol{w}) = \ell^{\mathrm{labeled}}(\boldsymbol{w}) + \lambda \cdot \ell^{\mathrm{unlabeled}}(\boldsymbol{w}), \tag{3}$$

where $\lambda$ control the weight on the unlabeled data. We introduces some notable works surrounding semi-supervised learning below:

- Najafi et al. (2019) examine the robust semi-supervised learning approach using distributionally robust optimization. Their method involves assigning soft labels to unlabeled data based on an adversarial loss and training such images with labeled ones. Their experiments demonstrate that the proposed algorithm enhances adversarially robust generalization across a variety of tasks.

- Uesato et al. (2019) investigate the robustness of a Gaussian mixture model proposed by Schmidt et al. (2018) and demonstrate that a self-training approach can effectively utilize unlabeled data to improve the model's adversarial robustness. They apply the self-training algorithm on the real-world image dataset CIFAR-10, combined with unlabeled Tiny Image dataset, and achieve state-of-the-art adversarial robustness. Their approach shows significant improvements in low-labeled data regimes by removing most of the labels from CIFAR-10 and SVHN datasets.

- Carmon et al. (2019) achieve comparable theoretical and empirical outcomes as (Uesato et al., 2019), while also providing a more comprehensive examination of additional facets. They demonstrate that by leveraging unlabeled data and robust self-training, the trained models can achieve enhanced certified robustness against all potential attacks. Additionally, they investigate the effects of various training components on the ultimate model performance, including the size of the unlabeled data.

- Zhai et al. (2019) leverage risk decomposition theorem by separating the expected robust risk into two components: stability and accuracy. The stability component measures the prediction stability in the presence of perturbations, and the accuracy component measures the standard classification accuracy. As the stability component does not require label information, it can be optimized using unlabeled data. Their study focuses on a specific Gaussian mixture problem inspired by (Schmidt et al., 2018), which shows that adversarial robust generalization can be almost as easy as standard generalization in supervised learning if a sufficiently large amount of unlabeled data is provided.

## 2.3 Robust generalization

An interesting characteristic of deep neutral networks (DNNs) is their ability to generalize well in practice Belkin et al. (2019). For the standard training setting, it is observed that test loss continues to decrease for long periods of training Nakkiran et al. (2020), thus the common practice is to train DNNs for as long as possible. However, in the case of adversarial training, further training past a certain point leads to a significant decrease in the robust training loss of the classifier, while increasing the robust test loss. Figure 1 depicts this phenomenon for adversarial training on CIFAR-10, where the robust test error initially drops but then increases after the first learning rate decay. This phenomenon is called "robust overfitting", which has shown strong resistance to standard regularization techniques such as $l_1$, $l_2$ regularization and data augmentation methods, and can be observed on various datasets, including SVHN, CIFAR-100, and ImageNet. Rice et al. (2020).

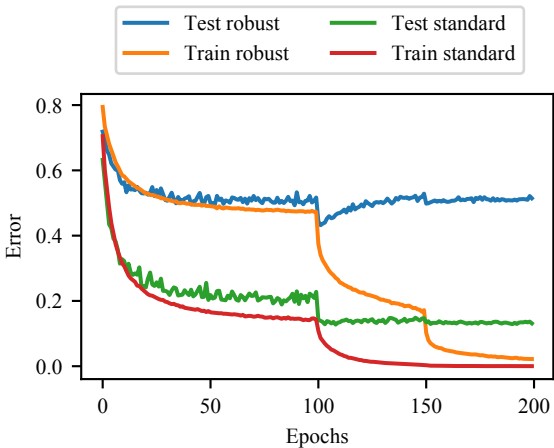

Figure 1: Adversarial training vs standard training curves. The depicted curves reveal "robust overfitting", wherein the adversarially trained model briefly achieves 43.2% robust error shortly after the first learning rate decay. Surprisingly, at this point, the adversarially trained model is actually more robust than it is at the end of training, where it only attains a 51.4% robust test error against a 10-step PGD adversary with an $\ell_\infty$ radius of $\epsilon = 8/255$. The learning rate is decayed at 100 and 150 epochs.

Schmidt et al. (2018) theorizes that robust generalization have a large sample complexity, which requires substantially larger dataset. Many subsequent works have empirically validated such claim, such as AT with semi-supervised learning Carmon et al. (2019); Zhai et al. (2019), robust local feature Song et al. (2020) and data interpolation Lee et al. (2020); Chen et al. (2021a). Chen et al. (2020) proposes to combine smoothing the logits via self-training and smoothing the weight via stochastic weight averaging to mitigate robust overfitting. Wu et al. (2020) emphasizes the connection of weight loss landscape and robust generalization gap, and suggests injecting the adversarial perturbations into both inputs and weights during AT to regularize the flatness of weight loss landscape. The intriguing property of robust overfitting has motivated great amount of study and investigation, but current works typically approach the phenomenon considering a DNN as a whole. In contrast, our work treats a DNN as a series of layers and reveals a strong connection between robust overfitting and the optimization of the latter layers, providing a novel perspective into better understanding the phenomenon.

## 3 Intriguing Properties of Robust Overfitting

In this section, we first investigate the layer-wise properties of robust overfitting by fixing model parameters in AT (Section 3.1). Based on our observations, we further propose a robust adversarial training (RAT) prototype to eliminate robust overfitting (Section 3.2). Finally, we design two different realizations for RAT to verify the effectiveness of the proposed method (Section 3.3).

### 3.1 Layer-wise Analysis of Robust Overfitting

Current works usually study the robust overfitting phenomenon considering the network as a single unit. However, features computed by different layers exhibit different properties, such as first-layer features are general and last-layer features are specific Yosinski et al. (2014). We hypothesize that different network layers have different effects on robust overfitting. To empirically verify the above hypothesis, we deliberately fix the parameters of the selected network layers, leaving them unoptimized during AT and observe the behavior of robust overfitting accordingly. Specifically, we considered ResNet-18 architecture as a composition of 4 main layers, corresponding to 4 Residual blocks. We then train multiple PreAct ResNet-18 networks on CIFAR-10 for 200 epochs using AT, each time selecting a set of network layers to have their parameter fixed.

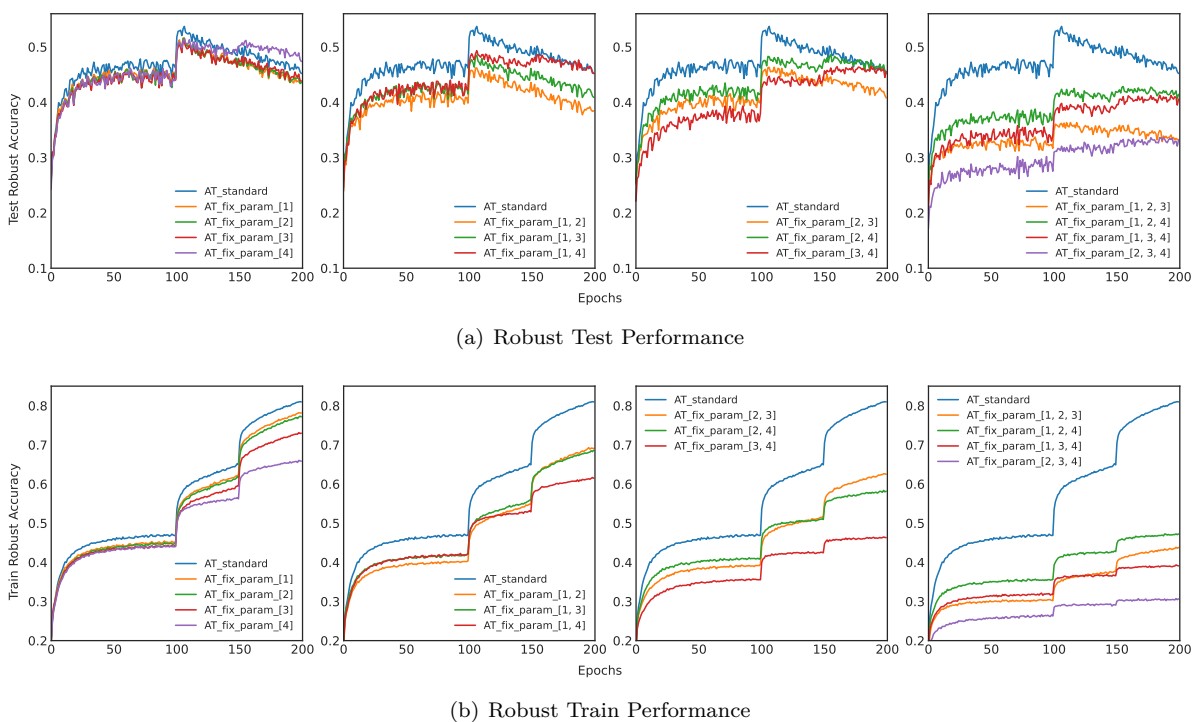

(a) Robust Test Performance

(b) Robust Train Performance

Figure 2: The robust train/test performance of adversarial training with different sets of network layers fixed. AT-fix-param[1,2] corresponds to fixing the parameters of layers 1 & 2 during AT

The robust test performance in figure 2(a) shows a consistent pattern. Robust overfitting is mitigated whenever we fix the parameters for layer 4 during AT, while any settings that do not fix the parameters for layer 4 result in a more severe gap between the best accuracy and the accuracy at the last epoch. For example, for settings such as AT-fix-param-[4], AT-fix-param-[1,4], AT-fix-param-[2,4] and AT-fix-param-[3,4], robust overfitting is significantly reduced. On the other hand, for settings such AT-fix-param-[1,2], AT-fix-param-[1,3] and AT-fix-param-[2,3], when we fix the parameters of various set of layers but allow for the optimization of layer 4, robust overfitting still widely exists. For extreme case like AT-fix-param-[1,2,3], where we fix the first three front layers and only allow for the optimization of that last layer 4, the gap between the best accuracy and the last accuracy is still obvious. This clearly indicates that the optimization of the latter layers present a strong correlation to the robust overfitting phenomenon. Note that this relationship can be observed across a variety of datasets, model architectures, and threat models (shown in Appendix A), indicating that it is a general property in adversarial training.

In many of these settings, robust overfitting is mitigated at the cost of robust accuracy. For example in AT-fix-param-[3,4], if we leave both layer 3 & 4 unoptimized, robust overfitting will practically disappear, but the peak performance is much worse compared to standard AT. When carefully examining the training performance in these settings shown in figure 2(b), we generally observe that the network capacity to fit adversarial data is strong when we fix the parameters for the front layers, but it gradually gets weaker as we try to fix the latter layers. For instance, AT-fix-param-[1] has the highest train robust accuracy, then comes AT-fix-param[2], AT-fix-param[3] and AT-fix-param[4]; AT-fix-param[1,2,3] has higher training accuracy than AT-fix-param[2,3,4]. This suggests fixing the latter layers' parameters can regularize the network better compared to fixing the front layers's parameters. In the subsequent sections, we will introduce methods that specifically regularize the optimization of the latter layers, so as to mitigate robust overfitting without tradeoffs in robustness. We will compare the impact on robust overfitting when applied such methods on the front layers vs the latter layers, further highlighting the importance of the latter layers in relation to robust overfitting.

## 3.2 A Prototype of RAT

As witnessed in Section 3.1, the optimization of AT in the latter layers is highly correlated to the existence of robust overfitting. To address this, we propose to train the network on adversarial data with some restrictions put onto the optimization of the latter layers, dubbed as *Robust Adversarial Training* (RAT). RAT adopts additional measures to regularize the optimization of the latter layers, and ensures that robust overfitting will not occur.

The RAT prototype is given in Algorithm 1. It runs as follows. We start with a base adversarial training algorithm $\mathcal{A}$. In Line 1-3, The inner maximization pass aims to maximize the loss via creating adversarial examples, and then the outer minimization pass updates the weight by minimizing the loss on adversarial data. Line 4 initiates a loop through all parts of the weight $w$ from the front layers to the latter layers. Line 5-9 then manipulate different parts of the weight based on its layer conditions. If the parts of the weight belong to the front layers ($\mathfrak{C}_{\text{front}}$), they will be kept intact. Otherwise, a weight update scheme $\mathfrak{S}$ is put onto the parts of the weight corresponding to the latter layers ($\mathfrak{C}_{\text{latter}}$). The role of $\mathfrak{S}$ is to apply some regularization on the latter layers' weight. Finally, the optimizer $\mathfrak{O}$ updates the model $f_w$ in Line 11.

Note that RAT is a general prototype where layer conditions $\mathfrak{C}_{\text{front}}$, $\mathfrak{C}_{\text{latter}}$ and weight adjustment strategy $\mathfrak{S}$ can be versatile. For example, based on the observations in Section 3.1, we treat the Res-Net architecture as a composition of 4 main layers, corresponding to 4 residual blocks, where $\mathfrak{C}_{\text{front}}$ indicates layer 1 & 2 and $\mathfrak{C}_{\text{latter}}$ indicates layer 3 & 4. $\mathfrak{S}$ can also represent various strategies that serves to regularize the optimization of the latter layers. In the section below, we will propose two different strategies $\mathfrak{S}$ in the implementations of RAT to demonstrate RAT's effectiveness.

---

**Algorithm 1** RAT-prototype (in a mini-batch).

---

**Require:** base adversarial training algorithm $\mathcal{A}$, optimizer $\mathfrak{O}$, network $f_w$, model parameter $w = \{w_1, w_2, ..., w_n\}$, training data $\mathcal{D} = \{(x_i, y_i)\}$, mini-batch $\mathcal{B}$, front and latter layer conditions $\mathfrak{C}_{\text{front}}$ and $\mathfrak{C}_{\text{latter}}$ for $f_w$, gradient adjustment strategy $\mathfrak{S}$

1: Sample a mini-batch $\mathcal{B} = \{(x_i, y_i)\}$ from $\mathcal{D}$
2: $\mathcal{B}' = \mathcal{A}.\text{inner\_maximization}(f_w, \mathcal{B})$
3: $\nabla_w \leftarrow \mathcal{A}.\text{outer\_minimization}(f_w, \ell_{\mathcal{B}'})$
4: **for** $i = 1, ..., n$ **do**
5:     **if** $\mathfrak{C}_{\text{front}}(w_i)$ **then**
6:         $\nabla_{w_i} \leftarrow \nabla_{w_i}$
7:     **else if** $\mathfrak{C}_{\text{latter}}(w_i)$ **then**
8:         $\nabla_{w_i} \leftarrow \mathfrak{S}(f_w, \mathcal{B}', \nabla_{w_i})$                         # adjust gradient
9:     **end if**
10: **end for**
11: $\mathfrak{O}.\text{step}(\nabla_w)$

---

## 3.3 Two Realizations of RAT

In this section, we will propose two different methods to put certain restrictions on the optimization of selected parts of the network, and then investigate the robust overfitting behavior upon applying such method to the front layers vs the latter layers. These methods showcase a clear relation between the optimization of the latter layers and robust generalization gap.

**RAT through enlarging learning rate**. In standard AT, the sudden increases in robust test performance appears to be closely related to the drops in the scheduled learning rate decay. We hypothesize that training AT without learning rate decays is sub-optimal, which can regularize the learning process of adversarial training. Comparison of the train/test performance between standard AT and AT without learning rate decay (AT-fix-lr-[1,2,3,4]) are shown in figure 3(b). Training performance of standard AT accelerates quickly right after the first learning rate drop, expanding the generalization gap with further training, whereas for AT without learning rate decay, training performance increases slowly and maintain a stable generalization gap. This suggests that AT optimized without learning rate decay has less capacity to fit adversarial data,

and thus provides the regularization needed to relieve robust overfitting. As our previous analysis suggests that the optimization of the latter layers is more important in mitigating robust overfitting, we propose using a fixed learning rate $= 0.1$ for optimizing the latter parts of the network while applying the piecewise decay learning rate for the former parts to close the robust generalization gap. We refer to this approach as a realization of RAT, namely $\mathrm{RAT_{LR}}$. Compared to standard AT, $\mathrm{RAT_{LR}}$ essentially enlarge the weight update step $\nabla_{w_i}$ along the latter parts of the gradients by 10 at the first learning rate decay and 100 at the second decay.

$$\nabla_{w_i} = \eta \nabla_{w_i}, \tag{4}$$

where $\eta$ is the amplification coefficient.

To demonstrate the effectiveness of $\mathrm{RAT_{LR}}$, we train multiple PreAct ResNet-18 networks on CIFAR-10 for 200 epochs using AT, each time selecting a set of network layers to have their learning rate fixed to 0.1 while maintaining the piece-wise learning rate schedule for other layers. Figure 3(a) validate our proposition. Robust overfitting is relieved for all settings that target layers that include layer 4 (AT-fix-lr-[4], AT-fix-lr-[1,4], AT-fix-lr-[2,4], etc.) while any settings that fix the learning rate of layers that exclude layer 4 do not reduce robust overfitting. Furthermore, all settings that fix the learning rate for both layer 3 & 4, including AT-fix-lr-[3,4], AT-fix-lr-[1,3,4], AT-fix-lr-[2,3,4] AT-fix-lr-[1,2,3,4] completely eliminate robust overfitting. The observations verify that regularizing the optimization of the latter layers by optimizing those layers without learning rate decays can prevent robust overfitting from occurring. An important observation is that $\mathrm{RAT_{LR}}$ (AT-fix-lr-[3,4]) can both overcome robust overfitting and achieve better robust test performance compared to the network using a fixed learning rate for all layers (AT-fix-lr-[1,2,3,4]). Examining the training performance between these two settings in figure 3(c), we find that $\mathrm{RAT_{LR}}$ exhibits a rapid rise in both robust and standard training performance immediately after the first learning rate decay similar to standard AT. The training performance of $\mathrm{RAT_{LR}}$ is able to benefit from the learning rate decay occurring at layer 1 & 2, making a notable improvement compared to AT-fix-lr-[1,2,3,4]. By training layers 3 & 4 without learning rate decays, we specifically put some restrictions on the optimization of only the latter parts of the network heavily responsible for robust overfitting, which can relieve robust overfitting without sacrificing too much performance. The experiment results provide another indication that the latter layers have stronger connections to robust overfitting than the front layers do, and regularizing the optimization of the latter layers from the perspective of learning rate can effectively solve robust overfitting.

**RAT through adversarial weight pertubation**. We continue to study the impact of different network layers to robust overfitting phenomenon from the perspective of adversarial weight perturbation (AWP). Wu et al. (2020) proposes AWP as a method to explicitly flatten weight loss landscape, by introducing adversarial perturbations into both inputs and weights during AT:

$$\min_w \max_{v \in V} \sum_i \max_{d(x_i, x_i') \leq \epsilon} \ell(f_{w+v}(x_i'), y_i), \tag{5}$$

where $v$ is the adversarial weight perturbation generated by maximizing the classification loss:

$$v = \nabla_w \sum_i \ell_i. \tag{6}$$

As AWP keeps injecting the worst-case perturbations on weight during training, it could also be viewed as a means to regularize the optimization of AT. In fact, the training of AWP exhibits a negative robust generalization gap, where robust training accuracy is in short of robust testing accuracy by a large margin, shown in figure 4(c). This indicates AWP put significant restrictions to the optimization of AT, introducing huge trade-offs to training performance. As our previous analysis suggests a strong correlation between robust overfitting and the optimization of the latter layers, we argue that the capacity to mitigate robust overfitting from AWP is mostly thanks to the perturbations occurring at latter layers' weight. As such, we propose to specifically apply AWP to the latter half of the network, and refer to this method as $\mathrm{RAT_{WP}}$. In essence, $\mathrm{RAT_{WP}}$ compute the adversarial weight perturbation $v_i$ under the layer condition $\mathfrak{C}_{\mathrm{latter}}(w_i)$, so that only the parts of the weight along the latter half of the network are perturbed.

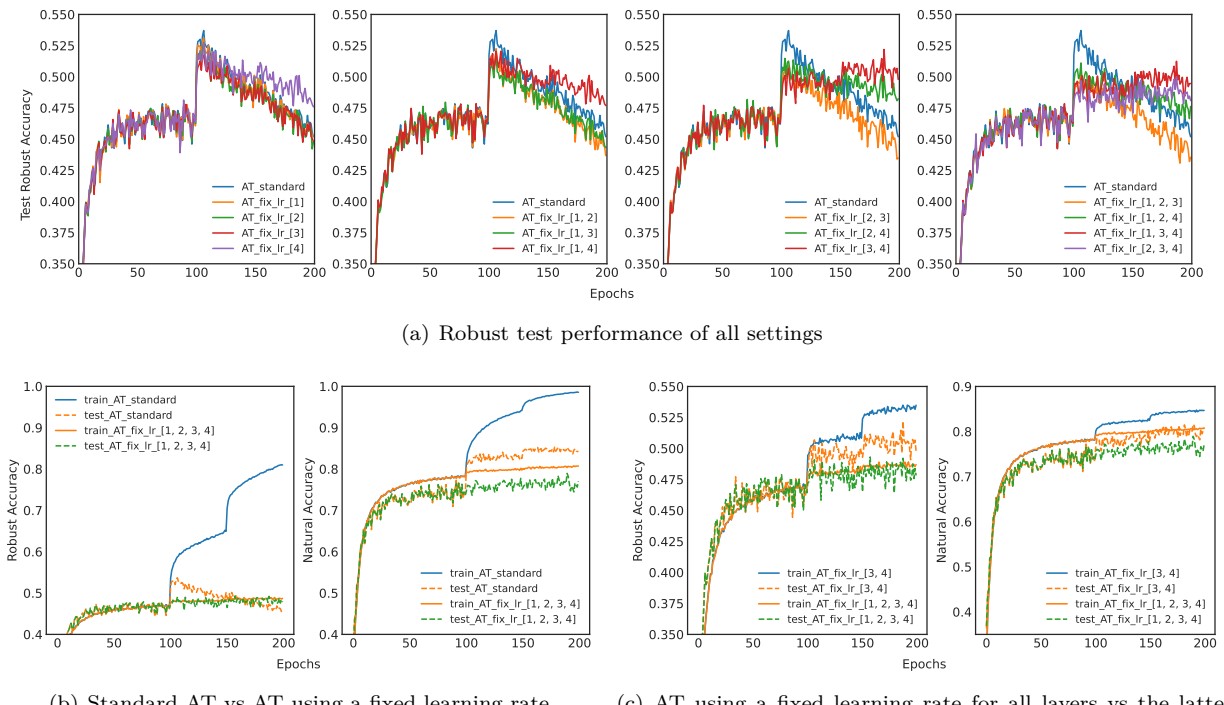

(a) Robust test performance of all settings

(b) Standard AT vs AT using a fixed learning rate

(c) AT using a fixed learning rate for all layers vs the latter layers

Figure 3: The train/test performance of adversarial training using a fixed learning rate for different sets of network layers. AT-fix-lr[1,2] corresponds to using a fixed learning rate for layers 1 & 2 during AT

$$\min_{\boldsymbol{w}=[w_1,...,w_i,...,w_n]} \max_{\boldsymbol{v}=[0,...,v_i,...0]\in V} \sum_i \max_{d(x_i,x_i')\le\epsilon} \ell(f_{\boldsymbol{w}+\boldsymbol{v}}(x_i'), y_i), \tag{7}$$

$$v_i = \nabla_{w_i} \sum_i \ell_i. \tag{8}$$

To prove the effectiveness of RAT$_{\text{WP}}$ , we train multiple PreAct ResNet-18 networks on CIFAR-10 for 200 epochs using AT, each time selecting a set of network layers to have their weight locally perturbed using AWP. As seen from figure 4(a), There are only 3 settings that can overcome robust overfitting, namely AT-AWP-[3,4], AT-AWP-[1,3,4] and AT-AWP-[2,3,4]. These settings share one key similarity: both layer 3&4 have their weight adversarially perturbed during AT. Simply applying AWP to any set of layers that exclude layers 3&4 is not sufficient to eliminate robust overfitting. This shows that AWP is effective in solving robust overfitting only when applied to both layer 3 and layer 4. Even when AWP is applied to the first 3 former layers out of 4 layers (AT-awp-[1,2,3]), robust overfitting still widely exists. In another word, it is essential for the adversarial weight perturbations to occur at the latter part of the network in order to mitigate robust overfitting. To examine this phenomenon in detail, we compare the training performance of AWP applied to front layers (represented by AT-AWP-[1,2,3]) vs AWP applied to latter layers (represented by AT-AWP-[3,4]), shown in figure 4(b). AWP applied in the front layers have a much better training performance than AWP applied in the latter layers. Furthermore, AWP applied to front layers reveals a positive robust generalization gap (training accuracy > testing accuracy) shortly after the first drop in learning rate, which continues to widen with further training. Conversely, AWP applied in the latter layers exhibits a negative robust generalization gap throughout most of the training, only converging to 0 after the second drop in learning rate. These differences demonstrate that worst-case perturbations, when injected into the latter layers' weights, have a more powerful impact in regularizing the optimization of AT. Consistent with our previous findings, AWP applied to the latter layers can be considered as an approach to regularize the optimization of AT in those layers, which successfully mitigates robust overfitting. This

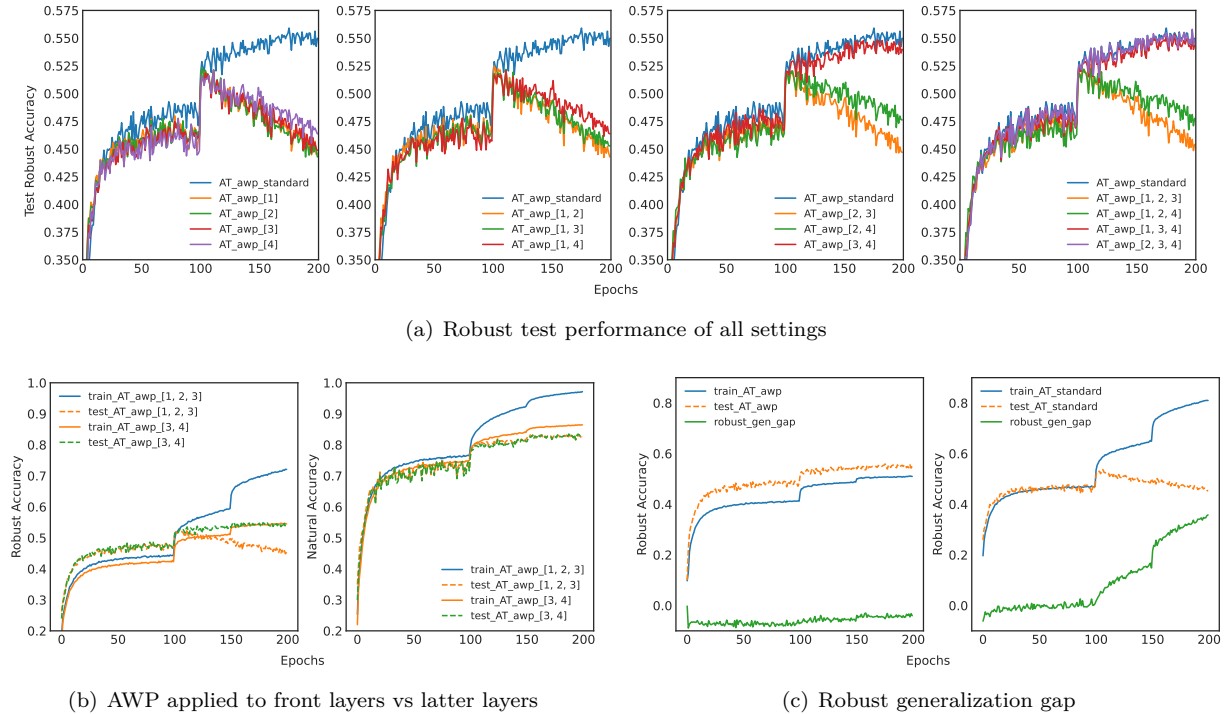

(a) Robust test performance of all settings

(b) AWP applied to front layers vs latter layers

(c) Robust generalization gap

Figure 4: The train/test performance of adversarial training when applying AWP for different sets of network layers. AT-AWP-[1,2] means only layer 1 & 2 have their weight perturbed using AWP

finding supports our analysis thus far, further demonstrating that regularizing the optimization of the latter layers is key to improving the robust generalization.

## 4 Experiment

In this section, we conduct extensive experiments to verify the effectiveness of $RAT_{LR}$ and $RAT_{WP}$. Details of the experiment settings and performance evaluation are introduced below.

### 4.1 Experimental Setup

We conduct extensive experiments on two realizations of RAT under two threat models ($L_\infty$ and $L_2$) across three benchmark datasets:

- CIFAR-10 (Krizhevsky et al., 2009). The CIFAR-10 dataset (Canadian Institute for Advanced Research, 10 classes) is a subset of the Tiny Images dataset and consists of 60000 32x32 color images. The images are labelled with one of 10 mutually exclusive classes: airplane, automobile (but not truck or pickup truck), bird, cat, deer, dog, frog, horse, ship, and truck (but not pickup truck). There are 6000 images per class with 5000 training and 1000 testing images per class.

- CIFAR100 (Krizhevsky et al., 2009). The CIFAR-100 dataset (Canadian Institute for Advanced Research, 100 classes) is a subset of the Tiny Images dataset and consists of 60000 32x32 color images. The 100 classes in the CIFAR-100 are grouped into 20 superclasses. There are 600 images per class. Each image comes with a "fine" label (the class to which it belongs) and a "coarse" label (the superclass to which it belongs). There are 500 training images and 100 testing images per class.

- SVHN (Netzer et al., 2011). Street View House Numbers (SVHN) is a digit classification benchmark dataset that contains 600,000 32×32 RGB images of printed digits (from 0 to 9) cropped from pictures

Table 1: Test robustness (%) on CIFAR10. We omit the standard deviations of 5 runs as they are very small ($< 0.6\%$).

| Network | Threat Model | Method | PGD-20 | | | AA | | |
|---|---|---|---|---|---|---|---|---|
| | | | Best | Last | Diff | Best | Last | Diff |
| PreAct ResNet-18 | $L_\infty$ | AT | 52.31 | 44.45 | 7.86 | 47.95 | 42.05 | 5.90 |
| | | $RAT_{LR}$ | 51.57 | 49.07 | 2.50 | 46.89 | 45.35 | 1.54 |
| | | $RAT_{WP}$ | **54.85** | **53.98** | **0.87** | **49.19** | **48.24** | **0.95** |
| | $L_2$ | AT | 69.27 | 65.86 | 3.41 | 67.65 | 64.64 | 3.01 |
| | | $RAT_{LR}$ | 68.97 | 68.21 | **0.76** | 64.26 | 63.44 | **0.82** |
| | | $RAT_{WP}$ | **70.77** | **69.49** | 1.28 | **68.29** | **67.11** | 1.18 |
| Wide ResNet-34-10 | $L_\infty$ | AT | 55.57 | 47.37 | 8.20 | 52.13 | 46.09 | 6.04 |
| | | $RAT_{LR}$ | 55.50 | 47.32 | 8.18 | 52.05 | 45.89 | 6.16 |
| | | $RAT_{WP}$ | **58.92** | **58.23** | **0.69** | **54.46** | **53.98** | **0.48** |
| | $L_2$ | AT | 70.57 | 68.99 | **1.58** | 69.44 | 66.92 | **2.52** |
| | | $RAT_{LR}$ | **71.91** | 68.94 | 2.96 | **70.53** | **67.90** | 2.63 |
| | | $RAT_{WP}$ | 71.31 | **69.19** | 2.12 | 70.12 | 67.35 | 2.77 |

Table 2: Test robustness (%) on CIFAR100. We omit the standard deviations of 5 runs as they are very small ($< 0.6\%$).

| Network | Threat Model | Method | PGD-20 | | | AA | | |
|---|---|---|---|---|---|---|---|---|
| | | | Best | Last | Diff | Best | Last | Diff |
| PreAct ResNet-18 | $L_\infty$ | AT | 28.07 | 21.24 | 6.83 | 23.61 | 18.41 | 5.20 |
| | | $RAT_{LR}$ | 26.57 | 26.18 | **0.39** | 21.77 | 21.22 | **0.55** |
| | | $RAT_{WP}$ | **30.91** | **30.42** | 0.49 | **25.52** | **24.57** | 1.05 |
| | $L_2$ | AT | 41.38 | 35.34 | 6.04 | 37.94 | 33.58 | 4.36 |
| | | $RAT_{LR}$ | 38.31 | 37.76 | 0.55 | 35.16 | 34.49 | **0.77** |
| | | $RAT_{WP}$ | **45.23** | **44.93** | **0.3** | **41.32** | **39.47** | 1.85 |
| Wide ResNet-34-10 | $L_\infty$ | AT | 30.74 | 24.89 | 5.85 | 26.98 | 23.07 | 3.91 |
| | | $RAT_{LR}$ | 30.57 | 23.53 | 7.04 | 26.72 | 22.53 | 4.19 |
| | | $RAT_{WP}$ | **30.81** | **25.46** | **5.35** | **27.11** | **23.56** | **3.55** |
| | $L_2$ | AT | 44.05 | 41.22 | 2.83 | 41.39 | 39.34 | 2.05 |
| | | $RAT_{LR}$ | 44.43 | 40.42 | 4.01 | 41.47 | 39.42 | 2.05 |
| | | $RAT_{WP}$ | **46.12** | **44.64** | **1.48** | **41.94** | **40.38** | **1.56** |

of house number plates. The cropped images are centered in the digit of interest, but nearby digits and other distractors are kept in the image. SVHN has three sets: training, testing sets and an extra set with 530,000 images that are less difficult and can be used for helping with the training process.

We use PreAct ResNet-18 He et al. (2016) and Wide ResNet-34-10 following the same hyperparameter settings for AT in Rice et al. (2020): for $L_\infty$ threat model, $\epsilon = 8/255$, step size is $1/255$ for SVHN, and $2/255$ for CIFAR-10 and CIFAR-100; for $L_2$ threat model, $\epsilon = 128/255$, step size is $15/255$ for all datasets. For training, all models are trained under 10-step PGD (PGD-10) attack for 200 epochs using SGD with momentum 0.9, weight decay $5 \times 10^{-4}$, and a piecewise learning rate schedule with an initial learning rate of 0.1. RAT models are decomposed into a series of 4 main layers, corresponding to 4 residual blocks of the ResNet architecture. For $RAT_{LR}$, learning rate for layer 3&4 are set to a fixed value of 0.1. For $RAT_{WP}$ leveraging AWP in layer 3&4, $\gamma = 1 \times 10^{-2}$. For testing, the robustness accuracy is evaluated under two different adversarial attacks, including 20-step PGD (PGD-20) and Auto Attack (AA) Croce & Hein (2020b). Auto Attack is considered the most reliable robustness evaluation to date, which is an ensemble of complementary attacks, consisting of three white-box attacks (APGD-CE (Croce & Hein, 2020b), APGD-

Table 3: Test robustness (%) on SVHN. We omit the standard deviations of 5 runs as they are very small ($< 0.6\%$).

| Network | Threat Model | Method | PGD-20 | | | AA | | |
|---|---|---|---|---|---|---|---|---|
| | | | Best | Last | Diff | Best | Last | Diff |
| PreAct ResNet-18 | $L_\infty$ | AT | 53.10 | 44.12 | 8.98 | 45.09 | 40.36 | **4.73** |
| | | $RAT_{LR}$ | 53.32 | 43.41 | 9.92 | 45.98 | 39.61 | 6.37 |
| | | $RAT_{WP}$ | **57.91** | **54.32** | **3.58** | **50.32** | **44.82** | 5.49 |
| | $L_2$ | AT | 66.29 | 64.73 | **1.55** | 63.55 | **60.14** | **3.41** |
| | | $RAT_{LR}$ | 66.47 | 62.10 | 4.36 | 62.44 | 58.72 | 3.72 |
| | | $RAT_{WP}$ | **71.66** | **65.68** | 5.98 | **65.17** | 59.64 | 5.53 |
| Wide ResNet-34-10 | $L_\infty$ | AT | 55.57 | 47.11 | 8.46 | 48.05 | 42.46 | 5.59 |
| | | $RAT_{LR}$ | 55.34 | 46.81 | 8.53 | 47.94 | 42.12 | 5.82 |
| | | $RAT_{WP}$ | **58.48** | **54.92** | **3.56** | **54.65** | **50.46** | **3.99** |
| | $L_2$ | AT | 67.19 | **65.08** | **2.11** | 62.58 | **60.86** | **1.72** |
| | | $RAT_{LR}$ | 67.50 | 64.24 | 3.27 | 62.79 | 59.94 | 2.85 |
| | | $RAT_{WP}$ | **69.07** | 64.76 | 4.31 | **63.12** | 59.57 | 3.55 |

Table 4: Performance (%) of RAT with other SOTA techniques on CIFAR-10 using PreAct ResNet-18 under $L_\infty$ threat model ($\epsilon = 8/288$)

| Method | PGD-20 | | |
|---|---|---|---|
| | Best | Last | Diff |
| AT | 53.10 | 44.12 | 8.98 |
| AWP | **55.39** | **54.73** | 0.66 |
| SWA | 52.14 | 51.53 | **0.61** |
| GAIRAT | 57.81 | 53.76 | 4.05 |
| RAT-LR | 51.57 | 49.07 | 2.5 |
| RAT-WP | 54.85 | 53.98 | 0.87 |

DLR (Croce & Hein, 2020b), and FAB (Croce & Hein, 2020a)) and a black-box attack (Square Attack (Andriushchenko et al., 2020))

## 4.2 Performance Evaluation

In this section, we present the experimental results of $RAT_{LR}$ and $RAT_{WP}$ across three benchmark datasets.

**CIFAR-10 Results.** The evaluation results on CIFAR10 dataset are summarized in Table 1, where "Best" is the highest test robustness achieved during training; "Last" is the test robustness at the last epoch checkpoint; "Diff" denotes the robust accuracy gap between the "Best" & "Last". It is observed that $RAT_{WP}$ generally achieves the best robust performance compared to $RAT_{LR}$ & standard AT. Regardless, both $RAT_{LR}$ and $RAT_{WP}$ tighten the robustness gaps by a significant margin, indicating they can effectively suppress robust overfitting.

**CIFAR-100 Results.** We also show the results on CIFAR100 dataset in Table 2. We observe similar performance like CIFAR10, where both $RAT_{LR}$ and $RAT_{WP}$ is able to significantly reduce the robustness gaps. For robustness improvement, $RAT_{WP}$ stands out to be the leading method. The results further verify the effectiveness of the proposed approach.

**SVHN Results.** The results on the SVHN dataset are shown in Table 3, where robustness gap are also narrowed down to a small margin by $RAT_{WP}$. SVHN dataset is a special case where $RAT_{LR}$ strategy does not improve robust overfitting. Unlike CIFAR10 and CIFAR100, learning rate decay in SVHN's training

does not have much connection to the sudden increases in robust test performance or the prevalence of robust overfitting, and hence makes $\text{RAT}_{\text{LR}}$ ineffective. Other than this, The improvement in robust generalization gaps can be witnessed in all cases, demonstrating the proposed approachs are generic and can be applied widely.

**Comparision with other SOTA methods.** Finally, we provide the performance comparison with some of the existing SOTA methods under PreAct-ResNet18 architecture, threat model and CIFAR-10 dataset in Table 4. These includes adversarial weight perturbation Wu et al. (2020), learned smoothening Chen et al. (2021b), instance reweighted Zhang et al. (2020). In general, our methods are comparable to most of the SOTA methods, achieving adequate robustness with low degree of robust overfitting.

## 5    Conclusion

In this paper, we investigate the effects of different network layers on robust overfitting and identify that robust overfitting is mainly driven by the optimization occurred at the latter layers. Following this, we propose a *robust adversarial training* (RAT) prototype to specifically hinder the optimization of the latter layers in the process of training adversarial network. The approach prevents the model from overfitting the latter parts of the network, which effectively eliminate robust overfitting of the network as a whole. We then further demonstrate two implementations of RAT: one locally uses a fixed learning rate for the latter layers and the other utilize adversarial weight perturbation for the latter layers. Extensive experiments show the effectiveness of both approaches, suggesting RAT is generic and can be applied across different network architectures, threat models and benchmark datasets to solve robust overfitting.

## 6    Future outlook

As much as our discovery has provided some interesting insights to the study of robust overfitting, we acknowledge that our work still have some limitations that future work can improve upon. Here we point out some them:

- To further explore the layer-wise properties of robust overfitting, we can broaden the extent of empirical study to consider more datasets (Tiny Imagenet, Standford Car, etc.) and network architectures (AlexNet, VGG, etc.).

- Ablation study on the effect of other factors such as different learning rate schedules (cyclic, cosine, annealed cosine etc.), the magnitude of the learning rate drop, the spacing of drop points could be thoroughly studied to showcase the generality of the properties.

- The proposed methods RAT-LR and RAT-AWP are kept fairly simple as a way to differentiate the properties of different layers, however, future work could attempt other strategies to regularize the weight of different network layers.

- The deeper layers/blocks are known to have many more parameters (especially for ResNet based models), thus a more detailed study of depth vs parameter count could shed light on the root cause of this layer-wise phenomenon.

Robust overfitting is a very interesting phenomenon in adversarial training that still lacks proper explanations on how and why it exists in the first place. We believe the layer-wise properties of robust overfitting could be the key piece, and addressing these points thoughtfully could provide important insights into the answers.

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

## A   More Evidences on the Layer-wise properties of Robust Overfitting

In this section, we provide more empirical experiments to showcase the layer-wise properties of robust overfitting across different datasets, model architectures and threat models. Specifically, we use two strategies mentioned in Section 3.3 to put restriction on the optimization of different network layers. We can always observe that there is no robust overfitting when we regularize the optimization of layers 3 and 4 (the latter layers), while robust overfitting is prevalent for other settings. These evidences further highlight the strong relation between robust overfitting and the optimization of the latter layers.

### A.1   Evidences across datasets

We show that the layer-wise properties of robust overfitting is universal across datasets on CIFAR-100 and SVHN. We adversarially train PreAct ResNet-18 under $l_\infty$ threat model on different datasets with the same settings as Section 3.3. The results are shown in Figure 5 and 6. Note that for SVHN, regularization strategy utilizing a fixed learning rate ($\text{RAT}_{\text{LR}}$) for does not improve robust overfitting (Figure 5). Unlike CIFAR10 and CIFAR100, SVHN's training overfits way before the first learning rate decay. Also, learning rate decay in SVHN's training does not have any relation to the sudden increases in robust test performance or the appearance of robust overfitting. Hence, SVHN dataset is a special case where $\text{RAT}_{\text{LR}}$ does not apply. For all other cases, robust overfitting is effectively eliminated by regularizing the optimization of layers 3 and 4.

## A.2   Evidences across threat models

We further demonstrate that the generality of layer-wise properties of robust overfitting by conducting experiments under $l_2$ threat model across datasets. The settings are the same as Section 3.3. The results are shown in Figure 7 and 8. Under $l_2$ threat model, except for SVHN dataset where regularization strategy utilizing a fixed learning rate ($\text{RAT}_{\text{LR}}$) does not apply, robust overfitting is effectively eliminated by regularizing the optimization of layers 3 and 4.

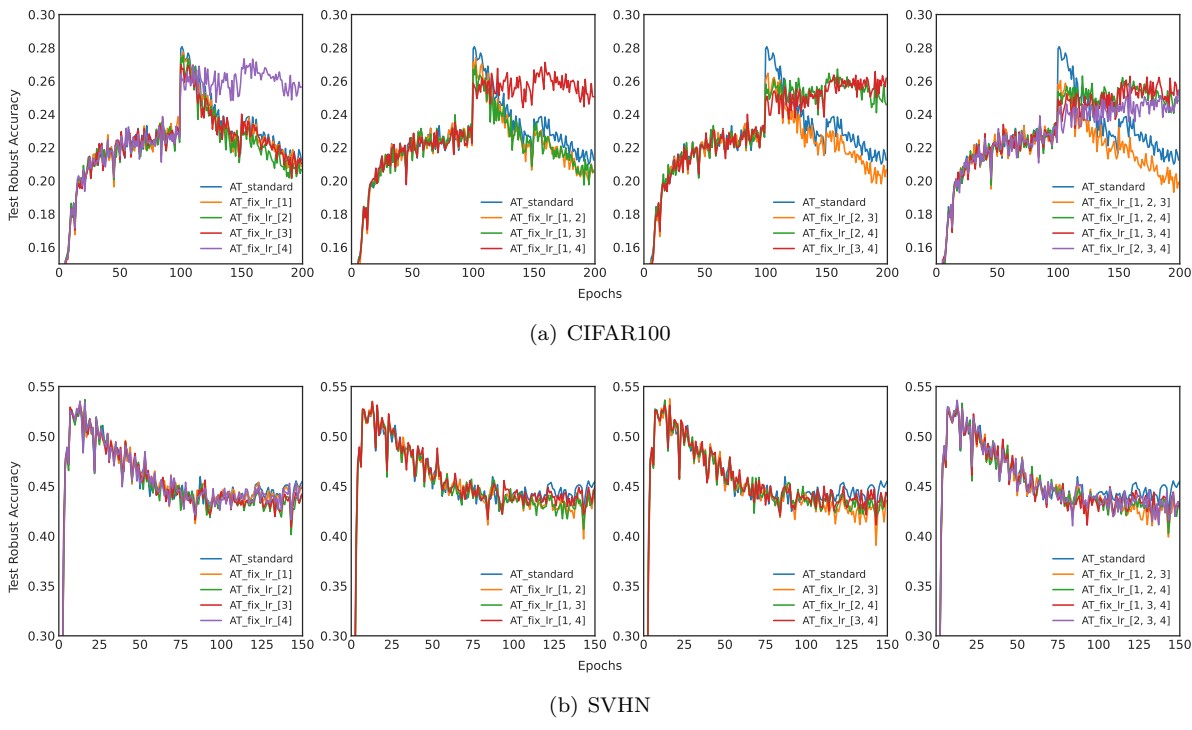

(a) CIFAR100

(b) SVHN

Figure 5: Robust test performance of adversarial training using a fixed learning rate for different sets of network layers, across datasets (CIFAR-100 and SVHN) under $l_\infty$ threat

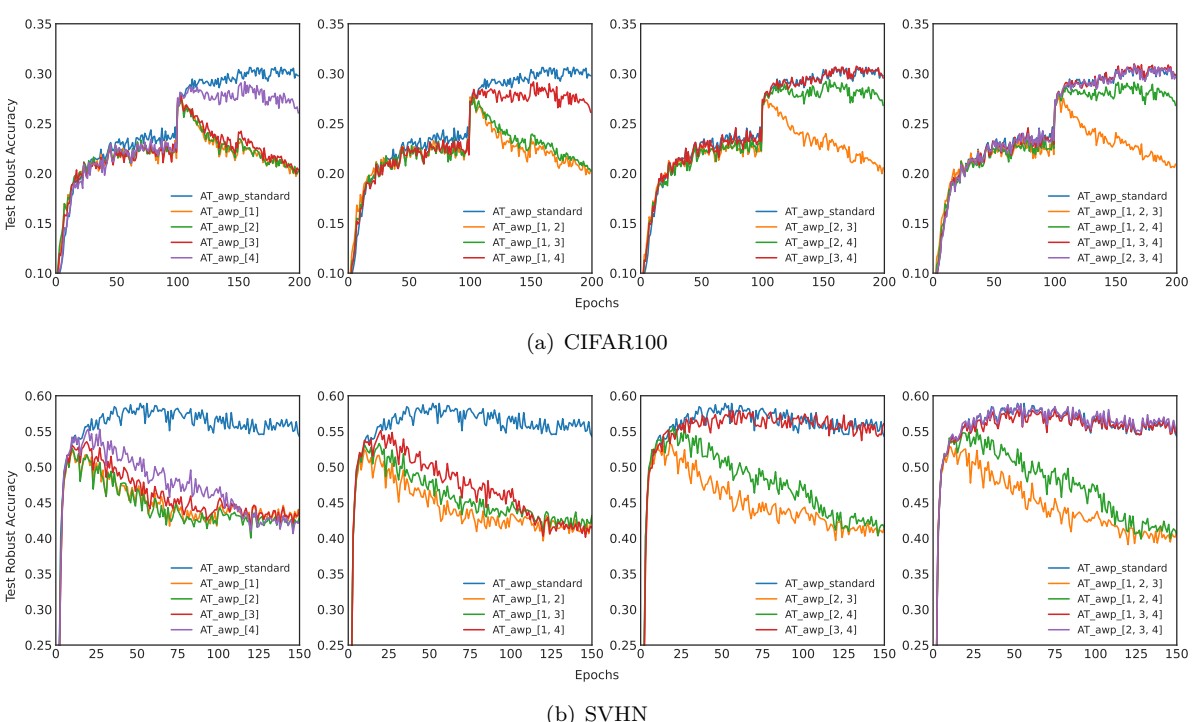

(a) CIFAR100

(b) SVHN

Figure 6: Robust test performance of adversarial training applying AWP for different sets of network layers, across datasets (CIFAR-100 and SVHN) under $l_\infty$ threat

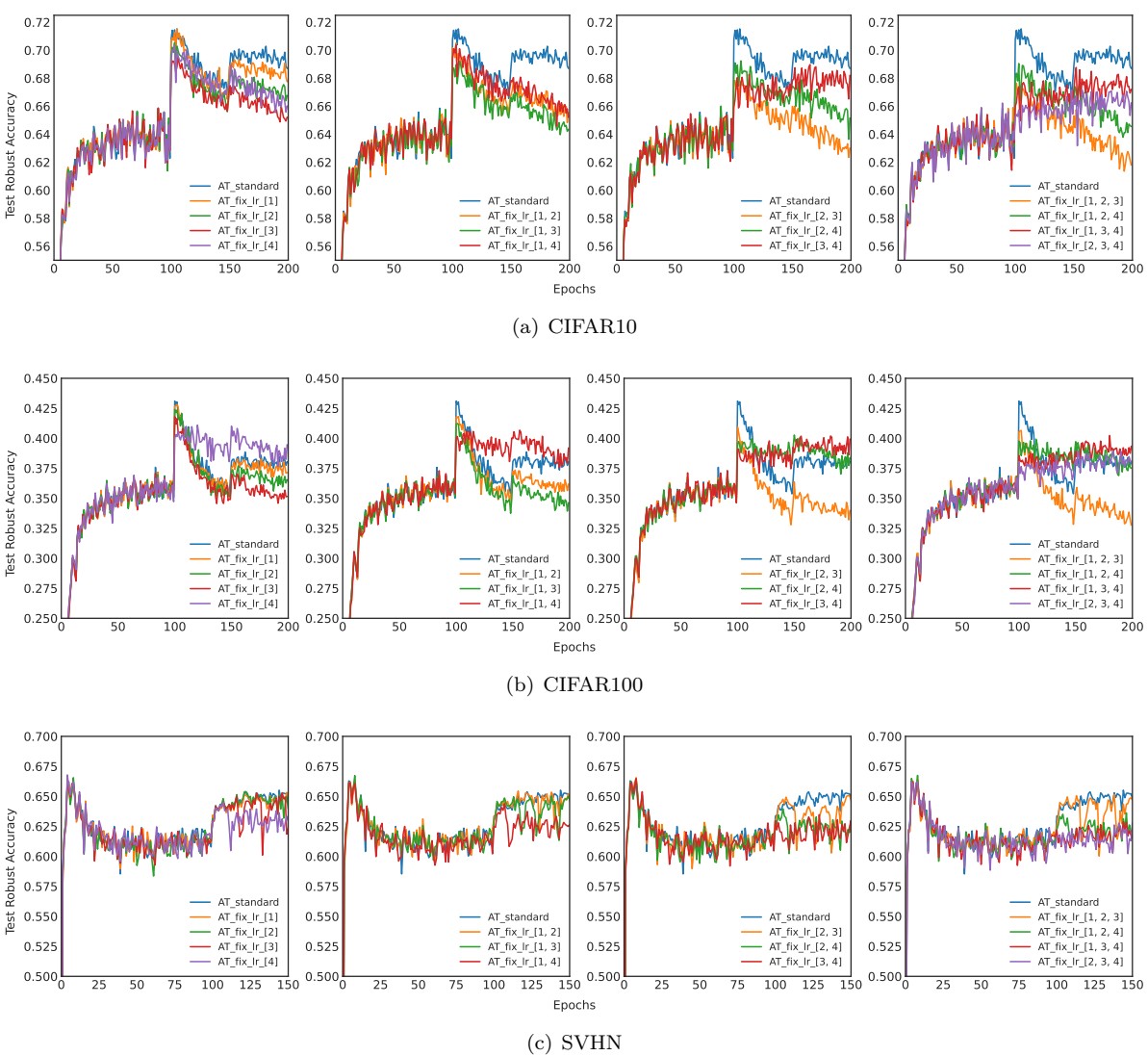

(a) CIFAR10

(b) CIFAR100

(c) SVHN

Figure 7: Robust test performance of adversarial training using a fixed learning rate for different sets of network layers, across datasets (CIFAR-10, CIFAR-100 and SVHN) under $l_2$ threat

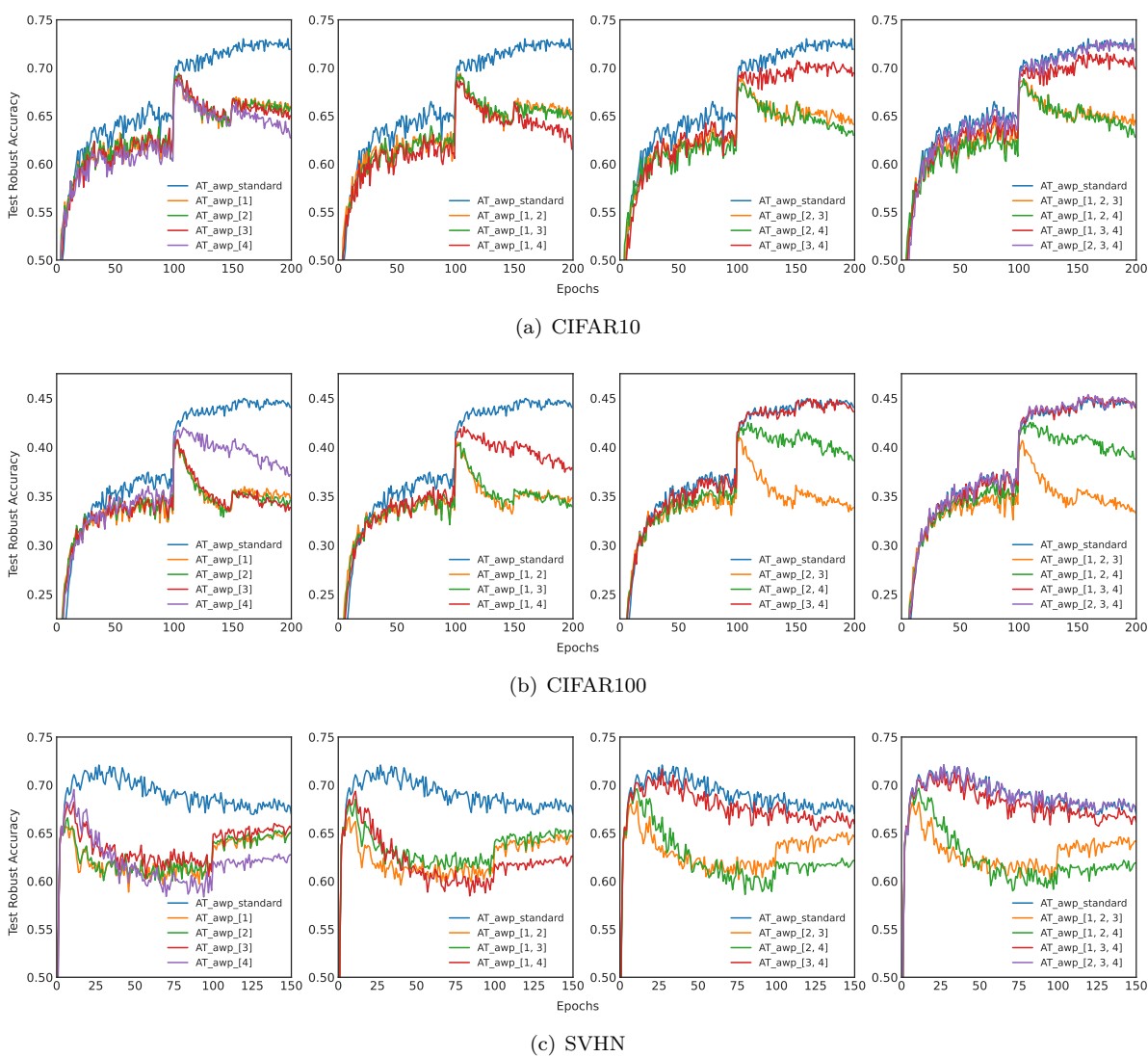

(a) CIFAR10

(b) CIFAR100

(c) SVHN

Figure 8: Robust test performance of adversarial training applying AWP for different sets of network layers, across datasets (CIFAR10, CIFAR-100 and SVHN) under $l_2$ threat

