# OpenReview forum: "On Intriguing Layer-Wise Properties of Robust Overfitting in Adversarial Training"
_TMLR — Rejected by TMLR_

### Review · Reviewer_rxRJ · 2023-03-18

**Summary Of Contributions:**

This paper studies how each layer of a neural network can affect adversarial training, specifically the robust overfitting phenomenon. This paper shows that layers of later stages of ResNet can have a larger impact. Based on this empirical observation, this paper proposed a method called robust adversarial training (RAT), implemented with layer-wise learning rate and weight perturbation.

**Audience:**

Yes

**Broader Impact Concerns:**

No ethical concerns.

**Claims And Evidence:**

Yes

**Requested Changes:**

Please address questions in the Strengths And Weaknesses section.

**Strengths And Weaknesses:**

---
Strengths

- The main observation that layers in later stages are responsible for robust overfitting is interesting.

- The proposed method RAT is fairly simple. The overall presentation of the paper is easy to follow.



---

Weaknesses

- Experiments were only conducted with ResNet and its variant. Is the same phenomenon can be observed with other architectures? Such as DenseNet, VGG or VisionTransformer? It's not clear if the such observation is generic or specific to ResNet.


- Existing work has shown an intriguing property of later stages of the neural network that they cause instability [1]. Is the overfitting related to the Lipschitz as well?

- As suggested by the AWP paper [2], AWP can mitigate overfitting. The proposed method, RAT-WP, essentially applies AWP to certain layers. In Table 4, it seems to apply AWP to the entire model is more beneficial, e.g. better performance (both Best and Last) and less overfitting (Diff).

- All experiments are based on standard adversarial training. Can RAT also improve other methods? Such as TRADES, MART and GAIRAT? Is the overfitting caused by the last few blocks generic to different adversarial training methods? Can RAT also improve existing SOTA methods? Based on the RobustBench, the RAT-LR is not comparable with existing methods.  RAT-WP will lower the performance compared to applying AWP to the entire model. I do not see any benefit from using RAT.

[1] Exploring architectural ingredients of adversarially robust deep neural networks. NeurIPS 2022 \
[2] Adversarial Weight Perturbation Helps Robust Generalization. NeurIPS 2020.

---

> ### Author Response · Authors · 2023-04-06
> **Response to Reviewer rxRJ**
>
> We sincerely thank you for the comprehensive comments on our paper and please find the answers to your questions below.
>
> **Q1:** Experiments were only conducted with ResNet and its variant. Is the same phenomenon can be observed with other architectures? Such as DenseNet, VGG or VisionTransformer? It's not clear if the such observation is generic or specific to ResNet
>
> **A1:** Since existing works related to robust overfitting in adversarial training mostly explore the phenomenon within the family of Resnet architectures, we also choose similar settings to make it easy to follow and do comparison between different work. We strongly believe that as long as robust overfitting exist in a specific architecture, this layer-wise property would also exist. Regardless, we agree that extending experiments to a broader set of architecture would further demonstrate the generality of this layer-wise property, and we definitely would do it on the future version of this work.
>
> **Q2:** Existing work has shown an intriguing property of later stages of the neural network that they cause instability [1]. Is the overfitting related to the Lipschitz as well
>
> **A2:** The work by Hanxun Huang has interestingly shown that reducing parameters at deeper layers improves both perturbation stability and lipschitzness.
> I think this has a very strong connection to our discovery, where regularizing the latter layers can reduce robust overfitting.
> Intuitively, we think that since the latter layers are closer to the output, they might be more sensitive to the supervisory signals coming from the adversarial loss compared to their former counterparts, which can have more impact on the overall lipschitzness.
> In this sense, lipschitzness and robust overfitting could be correlated to a certain degree.
>
> **Q3:** As suggested by the AWP paper [2], AWP can mitigate overfitting. The proposed method, RAT-WP, essentially applies AWP to certain layers. In Table 4, it seems to apply AWP to the entire model is more beneficial, e.g. better performance (both Best and Last) and less overfitting (Diff).
>
> **A3:** We agree that our methods RAT_WP (AWP applied to the latter layers) is not as efficient as AWP (AWP applied to all layers). However, our main contribution is the discovery of different properties of each network layer towards the robust overfitting.
> Our RAT_WP methods serve to highlight that the latter layers are more critical to robust overfitting, since AWP when not applied to the latter layers is totally ineffective. We do not see our methods as a SOTA solution to improving adversarial robustness. Our main focus is to demonstrate that constraining the optimization of the latter layers can significantly reduce overfitting.
> There could be better methods of achieving this goal, instead of training with a fixed learning rate or AWP applied to the latter layers. It is still an open question to design an optimal method that utilize this discovery, which could potentially improve upon our current methods.
>
> **Q4:** All experiments are based on standard adversarial training. Can RAT also improve other methods? Such as TRADES, MART and GAIRAT? Is the overfitting caused by the last few blocks generic to different adversarial training methods? Can RAT also improve existing SOTA methods? Based on the RobustBench, the RAT-LR is not comparable with existing methods. RAT-WP will lower the performance compared to applying AWP to the entire model. I do not see any benefit from using RAT.
>
> **A4:** As mentioned above, we do not see RAT_lr or RAT_wp as a SOTA solution or an complementary methods to other SOTA solutions. They mainly serves to demonstrate our discovery: regularizing the optimization of the latter layers can significantly reduce robust overfitting. We hope that future work could design a more optimal solution leveraging this insight, ones that can potentially produce new SOTA results or be complementary to exisiting ones.

---

### Review · Reviewer_2eDu · 2023-03-30

**Summary Of Contributions:**

This paper provides a diagnosis of robust overfitting on different network layers in adversarial training. Based on the observation that the optimization of the latter layers and the robust overfitting are highly correlated, the authors then propose a robust adversarial training (RAT) method with two implementations that optimizes the front parts of the network and applies additional measures to regularize the optimization of the latter parts. Extensive experiments demonstrate that RAT is able to eliminate robust overfitting and outperforms standard adversarial training in adversarial robustness.


**Audience:**

Yes

**Broader Impact Concerns:**

I don't have any concerns on the ethical implications of this work.

**Claims And Evidence:**

Yes

**Requested Changes:**

> In Figure 1 the authors use error as the y-axis, while in the rest of the paper, the y-axis is accuracy. I think it's better to make them consistent.

> Please refrain from only using color to distinguish curves as in figures, as it is not friendly to readers with color blindness.

> The authors claim that "Current works usually study the robust overfitting phenomenon considering the network as a single unit." in Section 3.1. I think the reference to this claim is missing.

> Equation (4) seems misleading. Should it be consistent with the notation in Line 8 of Algorithm 1?

> $\ell_i$ in equation (6) is not defined. Is it $\ell(x_i)$ or $\ell(x'_i)$? Also, the authors claim that $v$ is the worst-case perturbation. But in fact, according to (6), there is only one gradient step. I think it is better to provide more explanation.

> In Equation (8), the index for layers and index for samples are both $i$, which is misleading. Maybe use $j$ for the layer index?

> In Table 4, is $\epsilon= 8/255$?

**Strengths And Weaknesses:**

Strengths:
> Robust overfitting is a topic that is not well-studied in my opinion. It is also an interesting idea of fixing different layers during adversarial training to study their effect on robustness overfitting.


Weaknesses:

> From Figure 2 we see that fixing layer 4 achieves smaller robust overfitting but in $RAT_{LR}$ the authors increase the learning rate of the latter layers. Is this inconsistent with Figure 2?

> The writing of the paper is not very clear. Please see the Requested Changes.

> In the experiments, though we see a clear improvement in the test robustness (%), the numbers are still far from the SOTA results (e.g. CIFAR10 under AA and Linf: ~60%). I admit that this paper is improving vanilla AT, but AT is not strong enough compared to other more advanced methods. So if the authors can have some studies on applying RAT to the SOTA adversarial training methods and show similar improvement, the contribution is going to be much more significant.

---

> ### Author Response · Authors · 2023-04-05
> **Response to reviewer 2eDu**
>
> **Q1:** From Figure 2 we see that fixing layer 4 achieves smaller robust overfitting but in RAT_lr
> the authors increase the learning rate of the latter layers. Is this inconsistent with Figure 2?
>
> **A1:** From the empirical evidences in page 6, we learn that piecewise learning rate decays (learning rate divided by 10 at epoch 100th and by 100 at epoch 150th) seems to be the most efficient learning rate schedule for training AT.
> Thus, we demonstrate that training AT without learning rate decay has less capacity of fit adversarial data, which could act as a regularization scheme against robust overfitting.
> Increase the learning rate of the latter layers in this case means keeping a fixed learning rate without decay for the latter layers, which could produce similar effect like fixing layer 4 in figure 2.
>
> **Q2:** The performance are still far from the SOTA results
>
> **A2:** Our main contribution is the discovery of different properties of each network layer towards the robust overfitting phenomenon. Our methods serve to highlight that the latter layers are more sensitive to robust overfitting, and thus constraining its optimization can significantly reduce overfitting. There could be better methods of achieving this goal, instead of training with a fixed learning rate or AWP for latter layers. It is still an open question to design an optimal method that utilize this discovery, which could potentially improve upon our current method.
>
> **Q3**: In Figure 1 the authors use error as the y-axis, while in the rest of the paper, the y-axis is accuracy. I think it's better to make them consistent. Please refrain from only using color to distinguish curves as in figures, as it is not friendly to readers with color blindness.
>
> **A3**: We are very sorry for our lack of presentation in these cases. We will pay more attention on these when producing future work. We sincerely hope you do not find this too inconvenient to read.
>
> **Q4**: The authors claim that "Current works usually study the robust overfitting phenomenon considering the network as a single unit." in Section 3.1. I think the reference to this claim is missing.
>
> **A4**: We make this claim because among various work related to robust overfitting we have reviewed, none has actually consider studying it from the angle of separating the network layers.
> Regardless, we agree that without proper reference, we should refrain from making such claim. We thank you for pointing it out and we will be more careful in future work.
>
> **Q5**: Notation issues in equations
>
> **A5**: We are very sorry if our use of notations in equation are confusing or misleading. We try our best to follow the notations similar to other popular work in robust overfitting, such as in the work of Rice et al. (2020) (https://arxiv.org/pdf/2002.11569.pdf) and  Wu et al. (2020) (https://arxiv.org/pdf/2004.05884.pdf). We acknowledge that our use of notation might be hard to interpret, thus we have explained in detail how the method/algorithm behave after or before every equation.

---

> > ### Author Response · Authors · 2023-04-11
> >
> > Dear Reviewer 2eDu,
> >
> > As we only have one week left in this review process, if you have any further questions or additional clarification from us, please do not hesitate to let us know. We would be happy to provide it. Thank you!

---

### Review · Reviewer_JpQB · 2023-04-04

**Summary Of Contributions:**

The major contribution of this paper is two-folded:

1. The authors point out that the latter layers of the neural networks account more for adversarial overfitting seen in adversarial training.

2. The authors propose a generic framework (Algorithm 1 in the paper) that adaptively trains different layers to mitigate adversarial overfitting. In addition, the authors provide two concrete methods to regularize the training for parameters in the latter layers.

**Audience:**

Yes

**Broader Impact Concerns:**

Since no one has studied the adversarial overfitting phenomenon from the aspect of different layers, I think it would be beneficial to add a broader impact statement at the end of the paper to demonstrate the potential application of the findings in this paper.

**Claims And Evidence:**

Yes

**Requested Changes:**

I think the current manuscript misses several experiments and explanations to make the findings convincing, including but not limited to:

1. Experiments on more architectures, including ResNet of a different depth, networks without shortcut connection and ViT.

2. Stronger baselines and their combinations with layerwise adaptive training.

3. Explanations on why the latter layers contribute more to adversarial overfitting, either theoretically or empirically.

4. Proper way to use references.

5. (Optional) Experiments address the point 2 to point 4 in the minor issues of the "weakness part" in the last section.


**Strengths And Weaknesses:**

Strength:

To the best of my knowledge, it is the first work to study adversarial overfitting from the aspect of different layers in a model.

Weakness:

1. The architecture this work studies is ResNet, especially the one with 4 residual blocks, it is not clear whether or not the conclusions can generalize to other architectures, including ResNet with a different depth, networks without a shortcut connection and ViT.

2. Since the framework is generic, the authors should apply the proposed method to the SOTA robust learning method to conduct an ablation study. For example, the authors should include stronger baselines such as AWP in Table 1, 2, 3 and compare their performance with and without $RAT_{LR}$ or $RAT_{WP}$.

3. The authors do not provide any explanations on why the latter layers contribute more to adversarial overfitting. I think the findings would be more convincing with some reasonable explanation, including theoretical and empirical analyses.

Some minor issues:

1. The authors should use citation in the correct way. For example, in the first line after introduction, it should be "computer vision (He et al 2016)" instead of "computer vision He et al (2016)".

2. The authors say AWP has a negative generalization gap, which is strange. Do you use the same evaluation metric on the training and the test set? Is this just because you adversarially perturb the model parameter in the training set but not the test set?

3. Why not combine $RAT_{LR}$ and $RAT_{WP}$ together to evaluate its performance? i.e., use both large learning rate and adversarial weight perturbation for parameters in the latter layers.

4. Since there is almost no adversarial overfitting in the early phase of training, i.e., before you decay the learning rate, is it more reasonable that you apply $RAT_{LR}$ or $RAT_{WP}$ only after learning rate decay?

---

> ### Author Response · Authors · 2023-04-05
> **Response to reviewer JpQB**
>
> We sincerely thank you for the comprehensive comments on our paper and please find the answers to your questions below.
>
> **Q1:** The architecture this work studies is ResNet, especially the one with 4 residual blocks, it is not clear whether or not the conclusions can generalize to other architectures, including ResNet with a different depth, networks without a shortcut connection and ViT.
>
> **A1:** Since existing works related to robust overfitting in adversarial training mostly explore the phenomenon within the family of Resnet architectures, we also choose similar settings to make it easy to follow and do comparison between different work.
> We strongly believe that as long as robust overfitting exist in a specific architecture, this layer-wise property would also exist. Regardless,
> We agree that extending experiments to a broader set of architecture would further demonstrate the generality of this layer-wise property, and we definitely would do it on the future version of this work.
>
> **Q2:** Since the framework is generic, the authors should apply the proposed method to the SOTA robust learning method to conduct an ablation study. For example, the authors should include stronger baselines such as AWP in Table 1, 2, 3 and compare their performance with and without
> RAT_lr or RAT_wp.
>
> **A2:** We agree that an ablation study of applying the proposed method to other SOTA methods could reveal more about its effectiveness. We acknowledge this shortcoming and thank you for pointing it out.
> Since our main contribution is the discovery of different properties of each network layer towards robust overfitting, we focus more on highlighting this rather than trying to show our method would yield a SOTA result.
>
> **Q3:** The authors do not provide any explanations on why the latter layers contribute more to adversarial overfitting. I think the findings would be more convincing with some reasonable explanation, including theoretical and empirical analyses.
>
> **A3:** One intuition is that DNNs are often optimized with backpropagation using SGD, supervisory signals will gradually propagate through the whole network from latter layers to former layers. In this sense, the latter layers might be more sensitive to the supervisory signals coming from the adversarial loss compared to their former counterparts. In this work https://arxiv.org/pdf/2106.15853.pdf, the authors also discover that the latter layers in a DNN are
> much more sensitive to label noise, while their former counterparts are quite robust. If we somehow consider adversarial data as "input noise", the same dynamics from label noise can apply. This is purely our intuition without sufficient theoretical proofs, thus we decide not to include it in the work.
>
> **Q4:** The authors say AWP has a negative generalization gap, which is strange. Do you use the same evaluation metric on the training and the test set? Is this just because you adversarially perturb the model parameter in the training set but not the test set?
>
> **A4:** Yes we use the same evaluation metrics on both the training and the test set. We also find this quite strange. It seems by injecting worst-case perturbation to the weights, AWP make the training process very difficult, which causes the training performance to be quite low, even lower that its test performance.
>
> **Q5:** Why not combine RAT_lr and RAT_wp together to evaluate its performance? i.e., use both large learning rate and adversarial weight perturbation for parameters in the latter layers.
>
> **A5:** Yes, combining two methods might indeed yield better performance. However as mentioned before, we use two distinct methods to highlight this interesting layer-wise properties of robust overfitting. We has not yet attempted to fine-tune the techniques to produce SOTA result.
>
> **Q6:** Since there is almost no adversarial overfitting in the early phase of training, i.e., before you decay the learning rate, is it more reasonable that you apply RAT_lr or RAT_wp only after learning rate decay?
>
> **A6:** RAT_lr skips the learning rate decay part for latter layers, thus the effect of RAT_lr actually apply after the learning rate decay. For RAT_wp, yes, it is worth a try to see if we can squeeze out the performance. We thank you for the suggestion.

---

> > ### Author Response · Authors · 2023-04-11
> >
> > Dear Reviewer JpQB,
> >
> > As we only have one week left in this review process, if you have any further questions or requires additional clarification from us, please do not hesitate to let us know. We would be happy to provide it. Thank you!

---

### Decision · Action_Editors · 2023-05-09

**Recommendation:** Reject

**Comment:**

The reviewers acknowledge the interest of this work, but express concerns about its current scope. In particular, the reviewers would like the authors to
- analyze and validate their claims on more diverse network architectures;
- investigate the effect of other adversarial training techniques, beyond the standard one, on their conclusions.
Furthermore, the reviewers expressed several concerns about the presentation of the paper, which should be addressed.

As such, and considering that TMLR does not have "major revision" as potential recommendation, the AE recommends to reject this manuscript. However, should the authors strongly believe that they can convincingly address the reviewers' concerns, the AE would be willing to receive a revised version of the paper. This revised version would nonetheless go through another full round of reviews.


**Audience:**

This paper is in general of interest to the TMLR audience, but, as stated by the reviewers, the current scope, focusing on ResNet architectures and standard adversarial training, might be too narrow.

**Claims And Evidence:**

The reviewers acknowledge the interest of studying the problem of adversarial overfitting from the perspective of the different layers in a deep network. Although they find the evidence shown by the authors for ResNet models convincing, they express concerns about the generality of the claims to other network architectures and to other adversarial training strategies.